# Simultaneous Model Selection and Optimization through Parameter-free Stochastic Learning

**Francesco Orabona**[*]
Yahoo! Labs
New York, USA
francesco@orabona.com

## Abstract

Stochastic gradient descent algorithms for training linear and kernel predictors are gaining more and more importance, thanks to their scalability. While various methods have been proposed to speed up their convergence, the model selection phase is often ignored. In fact, in theoretical works most of the time assumptions are made, for example, on the prior knowledge of the norm of the optimal solution, while in the practical world validation methods remain the only viable approach. In this paper, we propose a new kernel-based stochastic gradient descent algorithm that performs model selection while training, with no parameters to tune, nor any form of cross-validation. The algorithm builds on recent advancement in online learning theory for unconstrained settings, to estimate over time the right regularization in a data-dependent way. Optimal rates of convergence are proved under standard smoothness assumptions on the target function as well as preliminary empirical results.

## 1 Introduction

Stochastic Gradient Descent (SGD) algorithms are gaining more and more importance in the Machine Learning community as efficient and scalable machine learning tools. There are two possible ways to use a SGD algorithm: to optimize a batch objective function, e.g. [23], or to directly optimize the generalization performance of a learning algorithm, in a stochastic approximation way [20]. The second use is the one we will consider in this paper. It allows learning over streams of data, coming Independent and Identically Distributed (IID) from a stochastic source. Moreover, it has been advocated that SGD theoretically yields the best generalization performance in a given amount of time compared to other more sophisticated optimization algorithms [6].

Yet, both in theory and in practice, the convergence rate of SGD for any finite training set critically depends on the step sizes used during training. In fact, often theoretical analysis assumes the use of optimal step sizes, rarely known in reality, and in practical applications wrong step sizes can result in arbitrarily bad performance. While in finite dimensional hypothesis spaces simple optimal strategies are known [2], in infinite dimensional spaces the only attempts to solve this problem achieve convergence only in the realizable case, e.g. [25], or assume prior knowledge of intrinsic (and unknown) characteristic of the problem [24, 29, 31, 33, 34]. The only known practical and theoretical way to achieve optimal rates in infinite Reproducing Kernel Hilbert Space (RKHS) is to use some form of cross-validation to select the step size that corresponds to a form of model selection [26, Chapter 7.4]. However, cross-validation techniques would result in a slower training procedure partially neglecting the advantage of the stochastic training. A notable exception is the algorithm in [21], that keeps the step size constant and the number of epochs on the training set acts as a regularizer. Yet, the number of epochs is decided through the use of a validation set [21].

---

[*]Work done mainly while at Toyota Technological Institute at Chicago.

Note that the situation is exactly the same in the batch setting where the regularization takes the role of the step size. Even in this case, optimal rates can be achieved only when the regularization is chosen in a problem dependent way [12, 17, 27, 32].

On a parallel route, the Online Convex Optimization (OCO) literature studies the possibility to learn in a scenario where the data are not IID [9, 36]. It turns out that this setting is strictly more difficult than the IID one and OCO algorithms can also be used to solve the corresponding stochastic problems [8]. The literature on OCO focuses on the adversarial nature of the problem and on various ways to achieve adaptivity to its unknown characteristics [1, 11, 14, 15].

This paper is in between these two different worlds: *We extend tools from OCO to design a novel stochastic parameter-free algorithm able to obtain optimal finite sample convergence bounds in infinite dimensional RKHS*. This new algorithm, called Parameter-free STOchastic Learning (PiSTOL), has the same complexity as the plain stochastic gradient descent procedure and implicitly achieves the model selection while training, with no parameters to tune nor the need for cross-validation. The core idea is to change the step sizes over time in a data-dependent way. As far as we know, this is the first algorithm of this kind to have provable optimal convergence rates.

The rest of the paper is organized as follows. After introducing some basic notations (Sec. 2), we will explain the basic intuition of the proposed method (Sec. 3). Next, in Sec. 4 we will describe the PiSTOL algorithm and its regret bounds in the adversarial setting and in Sec. 5 we will show its convergence results in the stochastic setting. The detailed discussion of related work is deferred to Sec. 6. Finally, we show some empirical results and draw the conclusions in Sec. 7.

## 2 Problem Setting and Definitions

Let $\mathcal{X} \subset \mathbb{R}^d$ a compact set and $\mathcal{H}_K$ the RKHS associated to a Mercer kernel $K : \mathcal{X} \times \mathcal{X} \to \mathbb{R}$ implementing the inner product $\langle \cdot, \cdot \rangle_K$ that satisfies the reproducing property, $\langle K(\boldsymbol{x}, \cdot), f(\cdot) \rangle_K = f(\boldsymbol{x})$. Without loss of generality, in the following we will always assume $\|k(\boldsymbol{x}_t, \cdot)\|_K \leq 1$.

Performance is measured w.r.t. a loss function $\ell : \mathbb{R} \to \mathbb{R}_+$. We will consider *L-Lipschitz* losses, that is $|\ell(x) - \ell(x')| \leq L|x - x'|$, $\forall x, x' \in \mathbb{R}$, and *H-smooth* losses, that is differentiable losses with the first derivative $H$-Lipschitz. Note that a loss can be both Lipschitz and smooth. A vector $\boldsymbol{x}$ is a subgradient of a convex function $\ell$ at $\boldsymbol{v}$ if $\ell(\boldsymbol{u}) - \ell(\boldsymbol{v}) \geq \langle \boldsymbol{u} - \boldsymbol{v}, \boldsymbol{x} \rangle$ for any $\boldsymbol{u}$ in the domain of $\ell$. The differential set of $\ell$ at $\boldsymbol{v}$, denoted by $\partial \ell(\boldsymbol{v})$, is the set of all the subgradients of $\ell$ at $\boldsymbol{v}$. $\mathbf{1}(\Phi)$ will denote the indicator function of a Boolean predicate $\Phi$.

In the OCO framework, at each round $t$ the algorithm receives a vector $\boldsymbol{x}_t \in \mathcal{X}$, picks a $f_t \in \mathcal{H}_K$, and pays $\ell_t(f_t(\boldsymbol{x}_t))$, where $\ell_t$ is a loss function. The aim of the algorithm is to minimize the *regret*, that is the difference between the cumulative loss of the algorithm, $\sum_{t=1}^T \ell_t(f_t(\boldsymbol{x}_t))$, and the cumulative loss of an arbitrary and fixed competitor $h \in \mathcal{H}_K$, $\sum_{t=1}^T \ell_t(h(\boldsymbol{x}_t))$.

For the statistical setting, let $\rho$ a fixed but unknown distribution on $\mathcal{X} \times \mathcal{Y}$, where $\mathcal{Y} = [-1, 1]$. A training set $\{\boldsymbol{x}_t, y_t\}_{t=1}^T$ will consist of samples drawn IID from $\rho$. Denote by $f_\rho(x) := \int_{\mathcal{Y}} y d\rho(y|x)$ the *regression function*, where $\rho(\cdot|x)$ is the conditional probability measure at $x$ induced by $\rho$. Denote by $\rho_{\mathcal{X}}$ the marginal probability measure on $\mathcal{X}$ and let $\mathcal{L}_{\rho_{\mathcal{X}}}^2$ be the space of square integrable functions with respect to $\rho_{\mathcal{X}}$, whose norm is denoted by $\|f\|_{\mathcal{L}_{\rho_{\mathcal{X}}}^2} := \sqrt{\int_{\mathcal{X}} f^2(x) d\rho_{\mathcal{X}}}$. Note that $f_\rho \in \mathcal{L}_{\rho_{\mathcal{X}}}^2$. Define the $\ell$-*risk* of $f$, as $\mathcal{E}^\ell(f) := \int_{\mathcal{X} \times \mathcal{Y}} \ell(y f(x)) d\rho$. Also, define $f_\rho^\ell(x) := \arg\min_{t \in \mathbb{R}} \int_{\mathcal{Y}} \ell(yt) d\rho(y|x)$, that gives the *optimal $\ell$-risk*, $\mathcal{E}^\ell(f_\rho^\ell) = \inf_{f \in \mathcal{L}_{\rho_{\mathcal{X}}}^2} \mathcal{E}^\ell(f)$. In the binary classification case, define the *misclassification risk* of $f$ as $\mathcal{R}(f) := P(y \neq \text{sign}(f(x)))$. The infimum of the misclassification risk over all measurable $f$ will be called *Bayes risk* and $f_c := \text{sign}(f_\rho)$, called the *Bayes classifier*, is such that $\mathcal{R}(f_c) = \inf_{f \in \mathcal{L}_{\rho_{\mathcal{X}}}^2} \mathcal{R}(f)$.

Let $L_K : \mathcal{L}_{\rho_{\mathcal{X}}}^2 \to \mathcal{H}_K$ the integral operator defined by $(L_K f)(x) = \int_{\mathcal{X}} K(x, x') f(x') d\rho_{\mathcal{X}}(x')$. There exists an orthonormal basis $\{\Phi_1, \Phi_2, \cdots\}$ of $\mathcal{L}_{\rho_{\mathcal{X}}}^2$ consisting of eigenfunctions of $L_K$ with corresponding non-negative eigenvalues $\{\lambda_1, \lambda_2, \cdots\}$ and the set $\{\lambda_i\}$ is finite or $\lambda_k \to 0$ when $k \to \infty$ [13, Theorem 4.7]. Since $K$ is a Mercer kernel, $L_K$ is compact and positive. Therefore, the fractional power operator $L_K^\beta$ is well defined for any $\beta \geq 0$. We indicate its range space by

| **Algorithm 1** Averaged SGD. | **Algorithm 2** The Kernel Perceptron. |
|---|---|
| **Parameters:** $\eta > 0$ | **Parameters:** None |
| **Initialize:** $f_1 = \mathbf{0} \in \mathcal{H}_K$ | **Initialize:** $f_1 = \mathbf{0} \in \mathcal{H}_K$ |
| **for** $t = 1, 2, \ldots$ **do** | **for** $t = 1, 2, \ldots$ **do** |
|    Receive input vector $\boldsymbol{x}_t \in \mathcal{X}$ |    Receive input vector $\boldsymbol{x}_t \in \mathcal{X}$ |
|    Predict with $\hat{y}_t = f_t(\boldsymbol{x}_t)$ |    Predict with $\hat{y}_t = \text{sign}(f_t(\boldsymbol{x}_t))$ |
|    Update $f_{t+1} = f_t + \eta y_t \ell'(y_t \hat{y}_t) k(\boldsymbol{x}_t, \cdot)$ |    Suffer loss $\mathbf{1}(\hat{y}_t \neq y_t)$ |
| **end for** |    Update $f_{t+1} = f_t + y_t \mathbf{1}(\hat{y}_t \neq y_t) k(\boldsymbol{x}_t, \cdot)$ |
| Return $\bar{f}_T = \frac{1}{T} \sum_{t=1}^{T} f_t$ | **end for** |

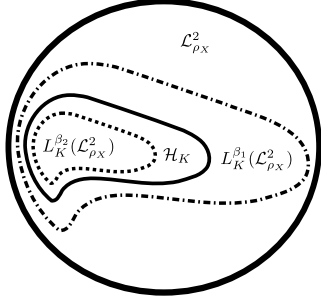

Figure 1: $\mathcal{L}^2_{\rho_{\mathcal{X}}}$, $\mathcal{H}_K$, and $L_K^\beta(\mathcal{L}^2_{\rho_{\mathcal{X}}})$ spaces, with $0 < \beta_1 < \frac{1}{2} < \beta_2$.

$$L_K^\beta(\mathcal{L}^2_{\rho_{\mathcal{X}}}) := \left\{ f = \sum_{i=1}^\infty a_i \Phi_i \; : \; \sum_{i:a_i \neq 0} a_i^2 \lambda_i^{-2\beta} < \infty \right\}. \quad (1)$$

By the Mercer's theorem, we have that $L_K^{\frac{1}{2}}(\mathcal{L}^2_{\rho_{\mathcal{X}}}) = \mathcal{H}_K$, that is every function $f \in \mathcal{H}_K$ can be written as $L_K^{\frac{1}{2}} g$ for some $g \in \mathcal{L}^2_{\rho_{\mathcal{X}}}$, with $\|f\|_K = \|g\|_{\mathcal{L}^2_{\rho_{\mathcal{X}}}}$. On the other hand, by definition of the orthonormal basis, $L_K^0(\mathcal{L}^2_{\rho_{\mathcal{X}}}) = \mathcal{L}^2_{\rho_{\mathcal{X}}}$. Thus, the smaller $\beta$ is, the bigger this space of the functions will be,[1] see Fig. 1. This space has a key role in our analysis. In particular, we will assume that $f_\rho^\ell \in L_K^\beta(\mathcal{L}^2_{\rho_{\mathcal{X}}})$ for $\beta > 0$, that is

$$\exists g \in \mathcal{L}^2_{\rho_{\mathcal{X}}} \; : \; f_\rho^\ell = L_K^\beta g. \quad (2)$$

## 3 A Gentle Start: ASGD, Optimal Step Sizes, and the Perceptron

Consider the square loss, $\ell(x) = (1 - x)^2$. We want to investigate the problem of training a predictor, $\bar{f}_T$, on the training set $\{\boldsymbol{x}_t, y_t\}_{t=1}^T$ in a stochastic way, using each sample only once, to have $\mathcal{E}^\ell(\bar{f}_T)$ converge to $\mathcal{E}^\ell(f_\rho^\ell)$. The Averaged Stochastic Gradient Descent (ASGD) in Algorithm 1 has been proposed as a fast stochastic algorithm to train predictors [35]. ASGD simply goes over all the samples once, updates the predictor with the gradients of the losses, and returns the averaged solution. For ASGD with constant step size $0 < \eta \leq \frac{1}{4}$, it is immediate to show[2] that

$$\mathbb{E}[\mathcal{E}^\ell(\bar{f}_T)] \leq \inf_{h \in \mathcal{H}_K} \mathcal{E}^\ell(h) + \|h\|_K^2 (\eta T)^{-1} + 4\eta. \quad (3)$$

This result shows the link between step size and regularization: In expectation, the $\ell$-risk of the averaged predictor will be close to the $\ell$-risk of the best *regularized* function in $\mathcal{H}_K$. Moreover, the amount of regularization depends on the step size used. From (3), one might be tempted to choose $\eta = \mathcal{O}(T^{-\frac{1}{2}})$. With this choice, when the number of samples goes to infinity, ASGD would converge to the performance of the best predictor in $\mathcal{H}_K$ at a rate of $\mathcal{O}(T^{-\frac{1}{2}})$, *only if* the infimum $\inf_{h \in \mathcal{H}_K} \mathcal{E}^\ell(h)$ is attained by a function in $\mathcal{H}_K$. Note that even with a *universal kernel* we only have $\mathcal{E}^\ell(f_\rho^\ell) = \inf_{h \in \mathcal{H}_K} \mathcal{E}^\ell(h)$ but there is no guarantee that the infimum is attained [26].

On the other hand, there is a vast, and often ignored, literature examining the general case when (2) holds [4, 7, 12, 17, 24, 27, 29, 31–34]. Under this assumption, this infimum is attained only when $\beta \geq \frac{1}{2}$, *yet it is possible to prove convergence for $\beta > 0$*. In fact, when (2) holds it is known that $\min_{h \in \mathcal{H}_K} \left[ \mathcal{E}^\ell(h) + \|h\|_K^2 (\eta T)^{-1} \right] - \mathcal{E}^\ell(f_\rho^\ell) = \mathcal{O}((\eta T)^{-2\beta})$ [13, Proposition 8.5]. Hence, it was observed in [33] that setting $\eta = \mathcal{O}(T^{-\frac{2\beta}{2\beta+1}})$ in (3), we obtain $\mathbb{E}[\mathcal{E}^\ell(\bar{f}_T)] - \mathcal{E}^\ell(f_\rho^\ell) = \mathcal{O}\left(T^{-\frac{2\beta}{2\beta+1}}\right)$,

that is the optimal rate [27, 33]. Hence, the setting $\eta = \mathcal{O}(T^{-\frac{1}{2}})$ is optimal only when $\beta = \frac{1}{2}$, that is $f_\rho^\ell \in \mathcal{H}_K$. In all the other cases, the convergence rate of ASGD to the optimal $\ell$-risk is suboptimal. Unfortunately, $\beta$ is typically unknown to the learner.

On the other hand, using the tools to design self-tuning algorithms, e.g. [1, 14], it may be possible to design an ASGD-like algorithm, able to self-tune its step size in a data-dependent way. Indeed, we would like an algorithm able to select the optimal step size in (3), that is

$$\mathbb{E}[\mathcal{E}^\ell(\bar{f}_T)] \leq \inf_{h \in \mathcal{H}_K} \mathcal{E}^\ell(h) + \min_{\eta > 0} \|h\|_K^2 (\eta T)^{-1} + 4\eta = \inf_{h \in \mathcal{H}_K} \mathcal{E}^\ell(h) + 4 \|h\|_K T^{-\frac{1}{2}}. \quad (4)$$

In the OCO setting, this would correspond to a regret bound of the form $\mathcal{O}(\|h\|_K T^{\frac{1}{2}})$. An algorithm that has this kind of guarantee is the Perceptron algorithm [22], see Algorithm 2. In fact, for the Perceptron it is possible to prove the following mistake bound [9]:

$$\text{Number of Mistakes} \leq \inf_{h \in \mathcal{H}_K} \sum_{t=1}^{T} \ell^h(y_t h(\boldsymbol{x}_t)) + \|h\|_K^2 + \|h\|_K \sqrt{\sum_{t=1}^{T} \ell^h(y_t h(\boldsymbol{x}_t))}, \quad (5)$$

where $\ell^h$ is the hinge loss, $\ell^h(x) = \max(1 - x, 0)$. The Perceptron algorithm is similar to SGD but its behavior is independent of the step size, hence, it can be thought as always using the optimal one. Unfortunately, we are not done yet: While (5) has the right form of the bound, it is not a regret bound, rather only a mistake bound, specific for binary classification. In fact, the performance of the competitor $h$ is measured with a different loss (hinge loss) than the performance of the algorithm (misclassification loss). For this asymmetry, the convergence when $\beta < \frac{1}{2}$ cannot be proved. Instead, we need an online algorithm whose regret bound scales as $\mathcal{O}(\|h\|_K T^{\frac{1}{2}})$, returns the averaged solution, and, thanks to the equality in (4), obtains a convergence rate which would depend on

$$\min_{\eta > 0} \|h\|_K^2 (\eta T)^{-1} + \eta. \quad (6)$$

Note that (6) has the same form of the expression in (3), but with a minimum over $\eta$. Hence, we can expect such algorithm to always have the optimal rate of convergence. In the next section, we will present an algorithm that has this guarantee.

## 4 PiSTOL: Parameter-free STOchastic Learning

In this section we describe the PiSTOL algorithm. The pseudo-code is in Algorithm 3. The algorithm builds on recent advancement in unconstrained online learning [16, 18, 28]. It is very similar to a SGD algorithm [35], the main difference being the computation of the solution based on the past gradients, in line 4. Note that the calculation of $\|g_t\|_K^2$ can be done incrementally, hence, the computational complexity is the same as ASGD in a RKHS, Algorithm 1, that is $\mathcal{O}(d)$ in $\mathbb{R}^d$ and $\mathcal{O}(t)$ in a RKHS. For the PiSTOL algorithm we have the following regret bound.

**Theorem 1.** *Assume that the losses $\ell_t$ are convex and $L$-Lipschitz. Let $a > 0$ such that $a \geq 2.25L$. Then, for any $h \in \mathcal{H}_K$, the following bound on the regret holds for the PiSTOL algorithm*

$$\sum_{t=1}^{T} [\ell_t(f_t(\boldsymbol{x}_t)) - \ell_t(h(\boldsymbol{x}_t))] \leq \|h\|_K \sqrt{2a \left(L + \sum_{t=1}^{T-1} |s_t|\right) \log \left(\frac{\|h\|_K \sqrt{aLT}}{b} + 1\right)}$$
$$+ b\phi \left(a^{-1} L\right) \log (1 + T),$$

*where $\phi(x) := \frac{x}{2} \exp \left(\frac{\exp\left(\frac{x}{2}\right)(x+1)+2}{1 - x \exp\left(\frac{x}{2}\right) - x}\right) \left(\exp \left(\frac{x}{2}\right) (x+1) + 2\right)$.*

This theorem shows that PiSTOL has the right dependency on $\|h\|_K$ and $T$ that was outlined in Sec. 3 and its regret bound is also optimal up to $\sqrt{\log \log T}$ terms [18]. Moreover, Theorem 1 improves on the results in [16, 18], obtaining an almost optimal regret that depends on the sum of the absolute values of the gradients, rather than on the time $T$. This is critical to obtain a tighter bound when the losses are $H$-smooth, as shown in the next Corollary.

**Algorithm 3** PiSTOL: Parameter-free STOchastic Learning.

1: **Parameters:** $a, b, L > 0$
2: **Initialize:** $g_0 = \mathbf{0} \in \mathcal{H}_K$, $\alpha_0 = aL$
3: **for** $t = 1, 2, \dots$ **do**
4:     Set $f_t = g_{t-1} \frac{b}{\alpha_{t-1}} \exp\left(\frac{\|g_{t-1}\|_K^2}{2\alpha_{t-1}}\right)$
5:     Receive input vector $\boldsymbol{x}_t \in \mathcal{X}$
6:     *Adversarial setting:* Suffer loss $\ell_t(f_t(\boldsymbol{x}_t))$
7:     Receive subgradient $s_t \in partial\ell_t(f_t(\boldsymbol{x}_t))$
8:     Update $g_t = g_{t-1} - s_t k(\boldsymbol{x}_t, \cdot)$ and $\alpha_t = \alpha_{t-1} + a|s_t| \|k(\boldsymbol{x}_t, \cdot)\|_K$
9: **end for**
10: *Statistical setting:* Return $\bar{f}_T = \frac{1}{T} \sum_{t=1}^{T} f_t$

---

**Corollary 1.** *Under the same assumptions of Theorem 1, if the losses $\ell_t$ are also $H$-smooth, then*[3]

$$\sum_{t=1}^{T} [\ell_t(f_t(\boldsymbol{x}_t)) - \ell_t(h(\boldsymbol{x}_t))] = \tilde{\mathcal{O}}\left( \max\left\{ \|h\|_K^{\frac{4}{3}} T^{\frac{1}{3}}, \|h\|_K T^{\frac{1}{4}} \left( \sum_{t=1}^{T} \ell_t(h(\boldsymbol{x}_t)) + 1 \right)^{\frac{1}{4}} \right\} \right).$$

This bound shows that, if the cumulative loss of the competitor is small, the regret can grow slower than $\sqrt{T}$. It is worse than the regret bounds for smooth losses in [9, 25] because when the cumulative loss of the competitor is equal to 0, the regret still grows as $\tilde{\mathcal{O}}\left( \|f\|_K^{\frac{4}{3}} T^{\frac{1}{3}} \right)$ instead of being constant. However, the PiSTOL algorithm does not require the prior knowledge of the norm of the competitor function $h$, as all the ones in [9, 25] do.

In [19], we also show a variant of PiSTOL for linear kernels with almost optimal learning rate for each coordinate. Contrary to other similar algorithms, e.g. [14], it is a truly parameter-free one.

## 5 Convergence Results for PiSTOL

In this section we will use the online-to-batch conversion to study the $\ell$-risk and the misclassification risk of the averaged solution of PiSTOL. We will also use the following definition: $\rho$ has *Tsybakov noise exponent* $q \geq 0$ [30] iff there exist $c_q > 0$ such that

$$P_X(\{x \in \mathcal{X} : -s \leq f_\rho(x) \leq s\}) \leq c_q s^q, \quad \forall s \in [0, 1]. \tag{7}$$

Setting $\alpha = \frac{q}{q+1} \in [0, 1]$, and $c_\alpha = c_q + 1$, condition (7) is equivalent [32, Lemma 6.1] to:

$$P_X(\text{sign}(f(x)) \neq f_c(x)) \leq c_\alpha (R(f) - R(f_\rho))^\alpha, \quad \forall f \in \mathcal{L}_{\rho_\mathcal{X}}^2. \tag{8}$$

These conditions allow for faster rates in relating the expected excess misclassification risk to the expected $\ell$-risk, as detailed in the following Lemma that is a special case of [3, Theorem 10].

**Lemma 1.** *Let $\ell : \mathbb{R} \to \mathbb{R}_+$ be a convex loss function, twice differentiable at 0, with $\ell'(0) < 0$, $\ell''(0) > 0$, and with the smallest zero in 1. Assume condition (8) is verified. Then for the averaged solution $\bar{f}_T$ returned by PiSTOL it holds*

$$\mathbb{E}[\mathcal{R}(\bar{f}_T)] - \mathcal{R}(f_c) \leq \left( 32 \frac{c_\alpha}{C} \left( \mathbb{E}[\mathcal{E}^\ell(\bar{f}_T)] - \mathcal{E}^\ell(f_\rho^\ell) \right) \right)^{\frac{1}{2-\alpha}}, \quad C = \min\left\{ -\ell'(0), \frac{(\ell'(0))^2}{\ell''(0)} \right\}.$$

The results in Sec. 4 give regret bounds over arbitrary sequences. We now assume to have a sequence of training samples $(\boldsymbol{x}_t, y_t)_{t=1}^T$ IID from $\rho$. We want to train a predictor from this data, that minimizes the $\ell$-risk. To obtain such predictor we employ a so-called online-to-batch conversion [8]. For a convex loss $\ell$, we just need to run an online algorithm over the sequence of data $(\boldsymbol{x}_t, y_t)_{t=1}^T$, using the losses $\ell_t(x) = \ell(y_t x)$, $\forall t = 1, \cdots, T$. The online algorithm will generate a sequence of solutions $f_t$ and the online-to-batch conversion can be obtained with a simple averaging of all the solutions, $\bar{f}_T = \frac{1}{T} \sum_{t=1}^T f_t$, as for ASGD. The average regret bound of the online algorithm becomes a convergence guarantee for the averaged solution [8]. Hence, for the averaged solution of PiSTOL, we have the following Corollary that is immediate from Corollary 1 and the results in [8].

**Corollary 2.** *Assume that the samples $(\boldsymbol{x}_t, y_t)_{t=1}^T$ are IID from $\rho$, and $\ell_t(x) = \ell(y_t x)$. Then, under the assumptions of Corollary 1, the averaged solution of PiSTOL satisfies*

$$\mathbb{E}[\mathcal{E}^\ell(\bar{f}_T)] \leq \inf_{h \in \mathcal{H}_K} \mathcal{E}^\ell(h) + \tilde{\mathcal{O}}\left(\max\left\{\|h\|_K^{\frac{4}{3}} T^{-\frac{2}{3}}, \|h\|_K T^{-\frac{3}{4}} \left(T\mathcal{E}^\ell(h) + 1\right)^{\frac{1}{4}}\right\}\right).$$

Hence, we have a $\tilde{\mathcal{O}}(T^{-\frac{2}{3}})$ convergence rate to the $\phi$-risk of the best predictor in $\mathcal{H}_K$, if the best predictor has $\phi$-risk equal to zero, and $\tilde{\mathcal{O}}(T^{-\frac{1}{2}})$ otherwise. Contrary to similar results in literature, e.g. [25], we do not have to restrict the infimum over a ball of fixed radius in $\mathcal{H}_K$ and our bounds depends on $\tilde{\mathcal{O}}(\|h\|_K)$ rather than $\mathcal{O}(\|h\|_K^2)$, e.g. [35]. The advantage of not restricting the competitor in a ball is clear: The performance is always close to the best function in $\mathcal{H}_K$, *regardless of its norm*. The logarithmic terms are exactly the price we pay for not knowing in advance the norm of the optimal solution. For binary classification using Lemma 1, we can also prove a $\tilde{\mathcal{O}}(T^{-\frac{1}{2(2-\alpha)}})$ bound on the excess misclassification risk in the realizable setting, that is if $f_\rho^\ell \in \mathcal{H}_K$.

It would be possible to obtain similar results with other algorithms, as the one in [25], using a doubling-trick approach [9]. However, this would result most likely in an algorithm not useful in any practical application. Moreover, the doubling-trick itself would not be trivial, for example the one used in [28] achieves a suboptimal regret and requires to start from scratch the learning over two different variables, further reducing its applicability in any real-world application.

As anticipated in Sec. 3, we now show that the dependency on $\tilde{\mathcal{O}}(\|h\|_K)$ rather than on $\mathcal{O}(\|h\|_K^2)$ gives us the optimal rates of convergence in the general case that $f_\rho^\ell \in L_K^\beta(\mathcal{L}_{\rho_X}^2)$, without the need to tune any parameter. This is our main result.

**Theorem 2.** *Assume that the samples $(\boldsymbol{x}_t, y_t)_{t=1}^T$ are IID from $\rho$, (2) holds for $\beta \leq \frac{1}{2}$, and $\ell_t(x) = \ell(y_t x)$. Then, under the assumptions of Corollary 1, the averaged solution of PiSTOL satisfies*

- *If $\beta \leq \frac{1}{3}$ then $\mathbb{E}[\mathcal{E}^\ell(\bar{f}_T)] - \mathcal{E}^\ell(f_\rho^\ell) \leq \tilde{\mathcal{O}}\left(\max\left\{(\mathcal{E}^\ell(f_\rho^\ell) + 1/T)^{\frac{\beta}{2\beta+1}} T^{-\frac{2\beta}{2\beta+1}}, T^{-\frac{2\beta}{\beta+1}}\right\}\right).$*

- *If $\frac{1}{3} < \beta \leq \frac{1}{2}$, then $\mathbb{E}[\mathcal{E}^\ell(\bar{f}_T)] - \mathcal{E}^\ell(f_\rho^\ell)$*

  $$\leq \tilde{\mathcal{O}}\left(\max\left\{(\mathcal{E}^\ell(f_\rho^\ell) + 1/T)^{\frac{\beta}{2\beta+1}} T^{-\frac{2\beta}{2\beta+1}}, (\mathcal{E}^\ell(f_\rho^\ell) + 1/T)^{\frac{3\beta-1}{4\beta}} T^{-\frac{1}{2}}, T^{-\frac{2\beta}{\beta+1}}\right\}\right).$$

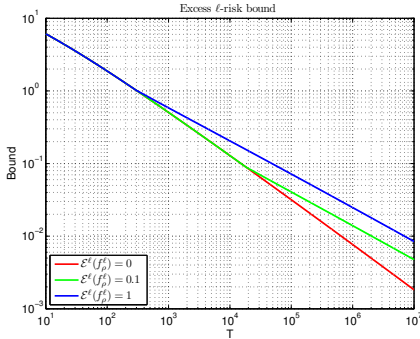

Figure 2: Upper bound on the excess $\ell$-risk of PiSTOL for $\beta = \frac{1}{2}$.

This theorem guarantees consistency w.r.t. the $\ell$-risk. We have that the rate of convergence to the optimal $\ell$-risk is $\tilde{\mathcal{O}}(T^{-\frac{3\beta}{2\beta+1}})$, if $\mathcal{E}^\ell(f_\rho^\ell) = 0$, and $\tilde{\mathcal{O}}(T^{-\frac{2\beta}{2\beta+1}})$ otherwise. However, for any finite $T$ the rate of convergence is $\tilde{\mathcal{O}}(T^{-\frac{2\beta}{\beta+1}})$ for any $T = \mathcal{O}(\mathcal{E}^\ell(f_\rho^\ell)^{-\frac{\beta+1}{2\beta}})$. In other words, we can expect a first regime at faster convergence, that saturates when the number of samples becomes big enough, see Fig. 2. This is particularly important because often in practical applications the features and the kernel are chosen to have good performance, meaning low optimal $\ell$-risk. Using Lemma 1, we have that the excess misclassification risk is $\tilde{\mathcal{O}}(T^{-\frac{2\beta}{(2\beta+1)(2-\alpha)}})$ if $\mathcal{E}^\ell(f_\rho^\ell) \neq 0$, and $\tilde{\mathcal{O}}(T^{-\frac{2\beta}{(\beta+1)(2-\alpha)}})$ if $\mathcal{E}^\ell(f_\rho^\ell) = 0$. It is also worth noting that, being the algorithm designed to work in the adversarial setting, we expect its performance to be robust to small deviations from the IID scenario.

Also, note that the guarantees of Corollary 2 and Theorem 2 hold *simultaneously*. Hence, the theoretical performance of PiSTOL is always better than both the ones of SGD with the step sizes tuned with the knowledge of $\beta$ or with the agnostic choice $\eta = \mathcal{O}(T^{-\frac{1}{2}})$. In [19], we also show another convergence result assuming a different smoothness condition.

Regarding the optimality of our results, lower bounds for the square loss are known [27] under assumption (2) and further assuming that the eigenvalues of $L_K$ have a polynomial decay, that is

$$(\lambda_i)_{i \in \mathbb{N}} \sim i^{-b}, \ b \geq 1. \tag{9}$$

Condition (9) can be interpreted as an effective dimension of the space. It always holds for $b = 1$ [27] and this is the condition we consider that is usually denoted as *capacity independent*, see the discussion in [21, 33]. In the capacity independent setting, the lower bound is $\mathcal{O}(T^{-\frac{2\beta}{2\beta+1}})$, that matches the asymptotic rates in Theorem 2, up to logarithmic terms. Even if we require the loss function to be Lipschitz and smooth, it is unlikely that different lower bounds can be proved in our setting. Note that the lower bounds are worst case w.r.t. $\mathcal{E}^\ell(f_\rho^\ell)$, hence they do not cover the case $\mathcal{E}^\ell(f_\rho^\ell) = 0$, where we get even better rates. Hence, the optimal regret bound of PiSTOL in Theorem 1 translates to an optimal convergence rate for its averaged solution, up to logarithmic terms, establishing a novel link between these two areas.

## 6  Related Work

The approach of stochastically minimizing the $\ell$-risk of the square loss in a RKHS has been pioneered by [24]. The rates were improved, but still suboptimal, in [34], with a general approach for locally Lipschitz loss functions in the origin. The optimal bounds, matching the ones we obtain for $\mathcal{E}^\ell(f_\rho^\ell) \neq 0$, were obtained for $\beta > 0$ in expectation by [33]. Their rates also hold for $\beta > \frac{1}{2}$, while our rates, as the ones in [27], saturate at $\beta = \frac{1}{2}$. In [29], high probability bounds were proved in the case that $\frac{1}{2} \leq \beta \leq 1$. Note that, while in the range $\beta \geq \frac{1}{2}$, that implies $f_\rho \in \mathcal{H}_K$, it is possible to prove high probability bounds [4, 7, 27, 29], the range $0 < \beta < \frac{1}{2}$ considered in this paper is very tricky, see the discussion in [27]. In this range no high probability bounds are known without additional assumptions. All the previous approaches require the knowledge of $\beta$, while our algorithm is parameter-free. Also, we obtain faster rates for the excess $\ell$-risk, when $\mathcal{E}^\ell(f_\rho^\ell) = 0$. Another important difference is that we can use any smooth and Lipschitz loss, useful for example to generate sparse solutions, while the optimal results in [29, 33] are specific for the square loss.

For finite dimensional spaces and self-concordant losses, an optimal parameter-free stochastic algorithm has been proposed in [2]. However, the convergence result seems specific to finite dimension.

The guarantees obtained from worst-case online algorithms, for example [25], have typically optimal convergence only w.r.t. the performance of the best in $\mathcal{H}_K$, see the discussion in [33]. Instead, all the guarantees on the misclassification loss w.r.t. a convex $\ell$-risk of a competitor, e.g. the Perceptron's guarantee, are inherently weaker than the presented ones. To see why, assume that the classifier returned by the algorithm after seeing $T$ samples is $f_T$, these bounds are of the form of $\mathcal{R}(f_T) \leq \mathcal{E}^\ell(h) + \mathcal{O}(T^{-\frac{1}{2}}(\|h\|_K^2 + 1))$. For simplicity, assume the use of the hinge loss so that easy calculations show that $f_\rho^\ell = f_c$ and $\mathcal{E}^\ell(f_\rho^\ell) = 2\mathcal{R}(f_c)$. Hence, even in the easy case that $f_c \in \mathcal{H}_K$, we have $\mathcal{R}(f_T) \leq 2\mathcal{R}(f_c) + \mathcal{O}(T^{-\frac{1}{2}}(\|f_c\|_K^2 + 1))$, i.e. no convergence to the Bayes risk.

In the batch setting, the same optimal rates were obtained by [4, 7] for the square loss, in high probability, for $\beta > \frac{1}{2}$. In [27], using an additional assumption on the infinity norm of the functions in $\mathcal{H}_K$, they give high probability bounds also in the range $0 < \beta \leq \frac{1}{2}$. The optimal tuning of the regularization parameter is achieved by cross-validation. Hence, we match the optimal rates of a batch algorithm, without the need to use validation methods.

In Sec. 3 we saw that the core idea to have the optimal rate was to have a classifier whose performance is close to the best regularized solution, where the regularizer is $\|h\|_K$. Changing the regularization term from the standard $\|h\|_K^2$ to $\|h\|_K^q$ with $q \geq 1$ is not new in the batch learning literature. It has been first proposed for classification by [5], and for regression by [17]. Note that, in both cases no computational methods to solve the optimization problem were proposed. Moreover, in [27] it was proved that *all* the regularizers of the form $\|h\|_K^q$ with $q \geq 1$ gives optimal convergence rates bound for the square loss, *given an appropriate setting of the regularization weight*. In particular, [27, Corollary 6] proves that, using the square loss and under assumptions (2) and (9), the optimal weight for the regularizer $\|h\|_K^q$ is $T^{-\frac{2\beta+q(1-\beta)}{2\beta+2/b}}$. This implies a very important consequence, not mentioned in that paper: In the the capacity independent setting, that is $b = 1$, if we use the regularizer $\|h\|_K$, *the optimal regularization weight is $T^{-\frac{1}{2}}$, independent of the exponent of the range space* (1) *where $f_\rho$ belongs*. Moreover, in the same paper it was argued that "From an algorithmic point of view however, q = 2 is currently the only feasible case, which in turn makes SVMs the method of choice". Indeed, in this paper we give a parameter-free efficient procedure to

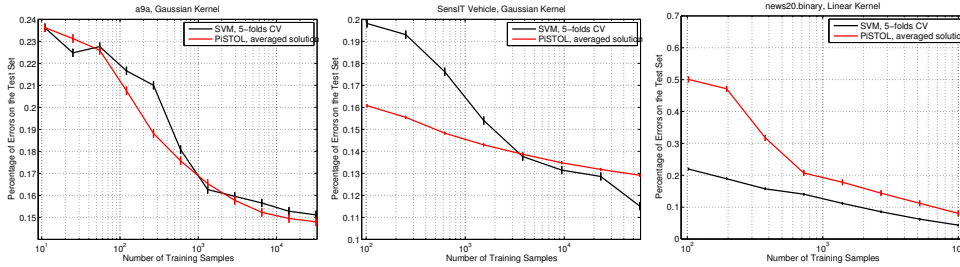

Figure 3: Average test errors and standard deviations of PiSTOL and SVM w.r.t. the number of training samples over 5 random permutations, on *a9a*, *SensIT Vehicle*, and *news20.binary*.

train predictors with smooth losses, that implicitly uses the $\|h\|_K$ regularizer. Thanks to this, the regularization parameter does not need to be set using prior knowledge of the problem.

## 7 Discussion

Borrowing from OCO and statistical learning theory tools, we have presented the first parameter-free stochastic learning algorithm that achieves optimal rates of convergence w.r.t. the smoothness of the optimal predictor. In particular, the algorithm does not require any validation method for the model selection, rather it automatically self-tunes in an online and data-dependent way.

Even if this is mainly a theoretical work, we believe that it might also have a big potential in the applied world. Hence, as a proof of concept on the potentiality of this method we have also run a few preliminary experiments, to compare the performance of PiSTOL to an SVM using 5-folds cross-validation to select the regularization weight parameter. The experiments were repeated with 5 random shuffles, showing the average and standard deviations over three datasets.[4] The latest version of LIBSVM was used to train the SVM [10]. We have that PiSTOL closely tracks the performance of the tuned SVM when a Gaussian kernel is used. Also, contrary to the common intuition, the stochastic approach of PiSTOL seems to have an advantage over the tuned SVM when the number of samples is small. Probably, cross-validation is a poor approximation of the generalization performance in that regime, while the small sample regime does not affect at all the analysis of PiSTOL. Note that in the case of News20, a linear kernel is used over the vectors of size $1355192$. The finite dimensional case is not covered by our theorems, still we see that PiSTOL seems to converge at the same rate of SVM, just with a worse constant. It is important to note that the total time the 5-folds cross-validation plus the training with the selected parameter for the SVM on 58000 samples of *SensIT Vehicle* takes $\sim 6.5$ hours, while our unoptimized Matlab implementation of PiSTOL less than 1 hour, $\sim 7$ times faster. The gains in speed are similar on the other two datasets.

This is the first work we know of in this line of research of stochastic adaptive algorithms for statistical learning, hence many questions are still open. In particular, it is not clear if high probability bounds can be obtained, as the empirical results hint, without additional hypothesis. Also, we only proved convergence w.r.t. the $\ell$-risk, however for $\beta \geq \frac{1}{2}$ we know that $f_\rho^\ell \in \mathcal{H}_K$, hence it would be possible to prove the stronger convergence results on $\left\| f_T - f_\rho^\ell \right\|_K$, e.g. [29]. Probably this would require a major change in the proof techniques used. Finally, it is not clear if the regret bound in Theorem 1 can be improved to depend on the squared gradients. This would result in a $\tilde{\mathcal{O}}(T^{-1})$ bound for the excess $\ell$-risk for smooth losses when $\mathcal{E}^\ell(f_\rho^\ell) = 0$ and $\beta = \frac{1}{2}$.

### Acknowledgments

I am thankful to Lorenzo Rosasco for introducing me to the beauty of the operator $L_K^\beta$ and to Brendan McMahan for fruitful discussions.

## Footnotes

[1]The case that $\beta < 1$ implicitly assumes that $\mathcal{H}_K$ is infinite dimensional. If $\mathcal{H}_K$ has finite dimension, $\beta$ is 0 or 1. See also the discussion in [27].

[2]The proofs of this statement and of all other presented results are in [19] .

[3]For brevity, the $\tilde{\mathcal{O}}$ notation hides polylogarithmic terms.

[4]Datasets available at http://www.csie.ntu.edu.tw/~cjlin/libsvmtools/datasets/. The precise details to replicate the experiments are in [19] .

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
