[Reviews · NeurIPS 2014]

Submitted by Assigned_Reviewer_19

The paper studies the interesting question on online (stochastic) gradient descent
in the unconstrained setting (sometime referred to non-strongly convexity or without
explicit regularization). In the earlier work [33], it was proved that the last
iterate of stochastic gradient descent with least-square loss in the unconstrained
setting actually converges with explicit convergence rate by appropriately choosing
the step sizes (or by a stopping early rule), which, however, needs to know the
smoothness of the regression function.

The paper proposed a kernel-based stochastic gradient descent algorithm without the
need to perform model selection, which requires the loss function and its gradient
are both Lipschitz. The proposed algorithm is mainly motivated by the recent
studies [15,16] which involves a data-dependent regularization. The main technical
proof for Theorem 1 involves elegant convex-analysis estimations, although the main
idea is similar to the standard telescope techniques for proving regret bounds in
online learning. The results of theorem 2 are also very interesting and are new
compared to the existing literature.

Some minor comments:

1. Can you prove the convergence of the last iterate of the proposed algorithm,
instead of the averaging one?

2. line 45: arbitrary -> arbitrarily

3. In line 153-159, you may wish to emphasize the results hold true for the least
square loss.

4. line 538-539: the claim that theorem 3 improves the results in [34] seems a bit
unfair, since the results hold for different loss functions and are based on the
assumptions on the smoothness of different target functions.

5. line 653-654: the second to last inequality is wrong

6. lines 678 and 710: z -> \kappa_t

7. The proof of lemma 7 does not need lemma 6. For the upper bound of $x$, one can
directly get it by applying the basic inequality: a (x+b)^alpha \le alpha (x+b) +
(1-alpha) a^{1-alpha}.
Summary: In summary, this is a very nice piece of work. The paper is also well written and
the related work and discussion are adequate.

Submitted by Assigned_Reviewer_22

Added post rebuttal: After considering the rebuttal and taking another look at the paper, I may have been a bit harsh in my criticism. I still think the paper relies a lot on techniques from previous papers, but the connections between them are indeed novel. I thank the author\s for the clarifications.

---------------------------------------------

This paper proposes a parameter-free algorithm for stochastic gradient descent of linear and kernel predictors. In particular, the step size is automatically determined and does not rely on the (initially unknown) norm of the competitor, nor on a doubling trick. The algorithm builds on recently developed online convex optimization algorithms with similar behavior, with the contributions being:
- Convergence bounds depending on the magnitude of the subgradients rather than the time horizon. In particular, for smooth losses, this leads to bounds which depend on the risk of the competitor, and interpolate between O(T^{-1/2}) and O(T^{-2/3}) rates.
- Convergence to the risk of the "Bayes-optimal" function (w.r.t. the loss function used), assuming this function lies in a function space with an appropriate spectral decay.
- Convergence bounds under a Tsybakov noise condition.
- A few experiments showing the competitiveness of the proposed algorithm and cross-validated SVM.

The question of how to avoid cross-validation and parameter tuning is certainly an interesting one and has practical implications. Moreover, the algorithm is simple and easy to implement. On the flip side, there has already been a string of recent works on online learning methods with such behavior, which can be readily applied to stochastic and batch problems (e.g. [14,16,25] which the paper cites). The proposed algorithm also builds on those methods, so the conceptual novelty is perhaps not as strong as the abstract and introduction seem to suggest. As far as I understand, the main improvement over them consists of the refined bound depending on the subgradient magnitudes, which for smooth losses lead to bounds depending on the loss of the competitor. This is an interesting contribution, although the bounds are probably suboptimal (e.g. when the loss is zero, we only get O(T^{-2/3}) rates, but with standard algorithms can get O(T^{-1}). A more minor issue is that it isn't clear if the refined bound and algorithmic change leads to anything practically noticeable (some experiments would be useful in that regard).

The results regarding convergence to the Bayes-optimal and Tsybakov noise condition are nice and technically non-trivial, but unless I missed something, appear to consist mostly of plugging the new bound into existing techniqus. Overall, I would say that the contributions are solid but a bit incremental in light of previous work.

In terms of presentation, the paper is mostly well-written, but I felt it was unnecessarily complicated at times. For example:
- In section 2, the definitions of the function spaces are very abstract, and would probably be hard to grasp for non-experts. Some specific examples would be very useful.
- Theorem 1 could be written in a much simpler way by choosing a specific value for "a", e.g. a=2.25L, and replacing \phi(x) by the resulting constant. I failed to see what additional insight is gained by keeping "a" as free parameter.

Technical and Other Comments
----------------------------
- The setting focuses on classification losses, which are a function of y*prediction. Can similar results be shown for regression or more general losses?

- Line 46: "While in finite hypothesis spaces simple optimal strategies are known, in infinite dimensional spaces..." - should "finite" be "finite dimensional"?

- Line 82: Since you focus on scalar-input loss functions, perhaps it's best to present things w.r.t. scalars rather than vectors, and use "sub-derivative" instead of "sub-gradient". For example, I was confused at first by algorithm 3, which talks about s_t as subgradients but are in fact used as scalars.

- Line 125: "By the Mercer's theorem"

- LIne 203: "We will present such algorithm"

- Line 315: "that is low optimal \ell-risk" -> "that has low optimal \ell-risk" (?)

- Line 408: "run few" -> "run a few"
Summary: The paper proposes a parameter-free algorithm for stochastic learning of linear and kernel predictors. This is an interesting direction, but the contribution is a bit incremental in light of recent such works.

Submitted by Assigned_Reviewer_26

This paper proposes a new online algorithm to learn supervised kernel machine. The required computation per iteration is quite simple, and the method has an adaptivity on the "complexity" of the true function in a sense that depending on the complexity, the learning rate adaptively achieves the minimax optimal rate.

I think the paper gives an interesting method. The analysis is also novel. Although the method is executed in an online manner, it shows adaptivity. This is a nice point.

A simple explanation of intuition of the algorithm might be helpful to the readers.

I am wondering the following point.
1. The method is not constructed to utilize the strong convexity of the loss function. Because of this, the learning rate is not depending on b, the decaying rate of the eigen-values. The local Rademacher complexity argument utilizes the strong convexity, and thus is able to improve the convergence rate depending on b. The dependence on beta is already interesting. But, I guess, if the method could involve b, the analysis is more interesting.

2. It is stated that the computational complexity is same as the plain SGD. However, the method requires to construct g_t and compute \|g_t\| at each iteration which should cost at least O(t). The current description is not clear about the actual computational cost. (I admit that the kernel method can't avoid O(T^2) computation. I am not criticizing that, but asking to clarify.)

===
After rebuttal:
- About the computational complexity
The authors claim that the computation per-iteration is O(1) because it is done incrementally from \|g_{t-1}\| and from \langle \g_{t-1}, x_t \rangle1}, x_t \rangle that is computed during the prediction step.
I see that the computation for the prediction is not included in the evaluation of the computational cost. My point was that the computation of \langle \g_{t-1}, x_t \rangle1}, x_t \rangle actually costs O(t), thus the cost per iteration should be O(t). I think it is better that this point is clearly written.
Summary: A nice theoretical paper which proposes a new online kernel method. The method seems practical and has a nice statistical convergence property.
Author Feedback
Author rebuttal: We thank the Reviewers for their very detailed and useful comments that will be useful in improving the paper.
We also thank Reviewer 19 and 26 for recognizing the potential high impact that we strongly believe this work has, from both a theoretical and practical point of view.

Reviewer 22: "The results regarding convergence to the Bayes-optimal and Tsybakov noise condition are nice and technically non-trivial, but unless I missed something, appear to consist mostly of plugging the new bound into existing techniques."

We respectfully but strongly disagree with Reviewer 22 on this point. Theorem 2 is our main result and the "existing techniques" the Reviewer mentions are actually first introduced in this very paper!
Specifically, the results in Theorem 2 are (mainly) proven thanks to the connection between optimal online learning regret bounds in unconstrained settings and optimal parameter-free SGD, sketched in Section 3 and in particular in the simple equation (4). The Reviewer seems to have missed the fact that this connection is shown for the *first time* in this paper. In fact, despite the simplicity of (4), no other paper contains even a hint at this solution. Indeed, the very opposite is true: As discussed in lines 371-399, the seminal paper by Steinwart et al. [24] strongly argued that regularizers of the form ||h||^q provides *no theoretical advantage* over the usual ||h||^2, apparently closing this research topic and making the result of Theorem 1 and the weaker results in [14,16,25] apparently useless. Here we prove the contrary.
Moreover, as also noted by Reviewer 19, we strongly believe that the result of Theorem 2 are "new" and "very interesting" with respect to the large body of literature on this topic.
The comments of the Reviewer made us realize that we did not stress enough this point, an oversight we will fix.

Reviewer 26: "Clarify complexity of the algorithm"

The calculation of \|g_t\| is just O(1). In fact, it should be done incrementally from \|g_{t-1}\| and from \langle \g_{t-1}, x_t \rangle that is computed during the prediction step. Hence, the overall computational complexity per iteration is exactly the same of SGD, with and without kernels. We will clarify it in the camera-ready version.

Reviewer 26: "how to utilize the strong convexity of the loss function?"

This is indeed a very good point and one of the important directions for future work. However, as far as we know, there are no SGD algorithms able to achieve an optimal dependency on “b”, not even with an oracle choice of the learning rate. We will add some comments addressing this in the discussion section.

Reviewer 19: “Can you prove the convergence of the last iterate of the proposed algorithm?”

Unfortunately no. The convergence of the last iterate seems currently out of reach with the analysis we used. For the same reason, even the convergence in norm K when beta>=1/2 seems difficult to achieve. We might have to switch to the analysis in [33] or find a way to fuse our approach with their approach.